# Ligand-Based Regulation of Dynamics and Reactivity of Hemoproteins

**DOI:** 10.3390/biom13040683

**Published:** 2023-04-17

**Authors:** Emily Samuela Turilli-Ghisolfi, Marta Lualdi, Mauro Fasano

**Affiliations:** Department of Science and High Technology, University of Insubria, 22100 Como, Italy

**Keywords:** heme, hemoproteins, globins, cytochromes, albumin, heme-albumin, hemopexin, nitrobindin, allostery, heme-based reactivity

## Abstract

Hemoproteins include several heme-binding proteins with distinct structure and function. The presence of the heme group confers specific reactivity and spectroscopic properties to hemoproteins. In this review, we provide an overview of five families of hemoproteins in terms of dynamics and reactivity. First, we describe how ligands modulate cooperativity and reactivity in globins, such as myoglobin and hemoglobin. Second, we move on to another family of hemoproteins devoted to electron transport, such as cytochromes. Later, we consider heme-based reactivity in hemopexin, the main heme-scavenging protein. Then, we focus on heme–albumin, a chronosteric hemoprotein with peculiar spectroscopic and enzymatic properties. Eventually, we analyze the reactivity and dynamics of the most recently discovered family of hemoproteins, i.e., nitrobindins.

## 1. Introduction

Heme (i.e., the iron(II) protoporphyrin IX complex) is a key cofactor of several proteins, conferring them oxygen-binding capability and redox reactivity with both physiological and pathological relevance [1,2,3]. Heme is found in the great majority of known life forms, and the organisms that utilize heme usually possess a complete pathway to synthesize the compound [4]. Five main families of heme-binding proteins exist, characterized by a peculiar heme coordination mode (Figure 1), and each is endowed with specific structure and function and spectroscopic properties (Table 1).

Myoglobin (Mb) and hemoglobin (Hb) were the first proteins whose structure was determined with X-ray crystallography, since the presence of the iron ion led to the solution of the phase problem [5]. Therefore, Mb and Hb gained a role of prototypical hemoproteins and were investigated with a variety of techniques to obtain a full kinetic and thermodynamic characterization of oxygen and ligand(s) binding [6,7]. To stress this aspect, Mb is defined as “the hydrogen atom of biology” [8]. In particular, Hb assumed the role of exemplary protein to describe cooperativity and allostery [7,9,10]. Related to Mb are neuroglobins and cytoglobins. These globins probably arose from an early vertebrate evolutionary divergence that led to moonlighting proteins (i.e., proteins with more than one function) [11,12,13].

A second class of heme proteins includes cytochromes, where the iron ion is in reversible equilibrium between the oxidized (Fe^3+^) and reduced (Fe^2+^) forms. Slight changes in the heme environment can finely tune the reduction potential to allow the transport of electrons across protein complexes [14]. In addition, cytochromes can bind ligands at the heme site either at the distal (canonic) side or at the proximal side of the heme group [15].

Another hemoprotein that plays a pivotal role in the control of the toxic side of heme reactivity is hemopexin. Indeed, heme can be present in plasma following intravascular hemolysis. Hemopexin is an acute-phase protein able to bind heme with high affinity, thus preventing the formation of reactive oxygen species catalyzed by free heme, and transfer it to the liver through CD91 receptor-mediated endocytosis [16,17]. In addition, it modulates anthracycline cardiac toxicity [18]. Hemopexin is also able to bind oxygen and other ligands, showing heme-based reactivity [19].

Before binding to hemopexin, plasma heme binds transiently to human serum albumin (HSA). This is a single-chain, all-alpha protein (585 amino acids) constituted by the repetition of three paralogous domains, probably arising from the reorganization of an ancestral nine-domain multipurpose protein still present in *cyclostoma* (lampreys), such as *Lethenteron japonicum* and *Petromyzon marinus* [20]. Although HSA is composed of a single polypeptide chain, it shows two main stable conformations called the N (neutral) and B (basic) forms. The transition between these two forms is cooperatively gated by the binding of fatty acid anions to seven main sites numbered from FA1 to FA7 [21]. Ferric heme (hemin) binds with high affinity (*K*_d_ = 1.0 × 10^−8^ M at pH = 7.0 and 24 °C [22]) to a D-shaped cavity that usually binds one fatty acid anion [23,24,25]. Consequently, heme–HSA acquires spectroscopic properties typical of heme proteins and heme-based reactivity [26].

Nitrobindin (Nb) indicates a family of mainly beta proteins that show a 10-stranded β-barrel resembling that of lipocalins and fatty acid binding proteins (FABPs) [27]. However, they appear to be an independent ubiquitous family of heme-binding proteins with respect to lipocalins [28,29,30]. All the Nb-like proteins display conserved residues in the pocket involved in the coordination and recognition of the heme–Fe(III). The crystallographic structure of the human Nb homolog (THAP4) showed a partial exposure to the solvent of the heme ring, suggesting a heme-based reactivity [29,31].

**Table 1 biomolecules-13-00683-t001:** Main spectroscopic properties of representatives of the hemoprotein classes described in this review.

Protein	UV–Vis	Resonance Raman	Fe Coordination Mode and Spin State	References
Sperm whale deoxy-myoglobin(II)	Soret band at 434 nm; Q band at 556 nm	1558 cm^−1^; 1341 cm^−1^, 1356 cm^−1^ (Fermi doublet); 1010 cm^−1^	5cHS	[32,33,34,35]
Human deoxy-Hemoglobin(II)	Soret band at 411 nm; Q band at 535 nm	1607 cm^−1^; 1552 cm^−1^; 1473 cm^−1^	5cHS	[32,36,37]
Human hemoglobin(III)	Soret band at 405 nm; several weaker Q bands at 450–650 nm	1639 cm^−1^; 1585 cm^−1^; 1372 cm^−1^	6cHS	[38,39]
Horse heart Cytochrome c(II)	Soret band at 415 nm; Q_1_ band at 520 nm; Q_0_ band at 550 nm	1361 cm^−1^; 1491 cm^−1^; 1592 cm^−1^; 1621 cm^−1^; 1546 cm^−1^	6cLS	[40,41,42]
Horse heart Cytochrome c(III)	Soret band at 409 nm; Q band at 529 nm; CT band at 695 nm	1372 cm^−1^; 1502 cm^−1^; 1584 cm^−1^; 1635 cm^−1^; 1560 cm^−1^	6cLS	[43,44]
Rabbit serum hemopexin(II)	Soret band at 426 nm; Q_1_ band at 526 nm; Q_0_ band at 556 nm	Not available	6cLS	[45,46]
Rabbit serum hemopexin(III)	Soret band at 413 nm; Q_1_ band at 530 nm; Q_0_ band at 563 nm	Not available	6cLS	[46,47]
Human serum albumin(II)	Soret band at 418 nm; shoulder at 405 nm; Q_1_ band at 536 nm; Q_0_ band at 572 nm	5cHS: 1358 cm^−1^; 1472 cm^−1^; 1557 cm^−1^; 1602 cm^−1^ 4cIS: 1370 cm^−1^; 1502 cm^−1^; 1580 cm^−1^; 1635 cm^−1^	Mixture of 5cHS and 4cIS	[48,49]
Human serum albumin(III)	Soret band at 404 nm; Q_1_ band at 501 nm; Q_0_ band at 533 nm; CT1 band at 622 nm	1493 cm^−1^; 1568 cm^−1^; ν_10_ 1624 cm^−1^	5cHS	[48]
*Homo sapiens* nitrobindin(III)	Soret band at 407 nm; CT1 band at 631 nm	ν_3_ 1487 cm^−1^; ν_2_ 1561 cm^−1^; ν_10_ 1611 cm^−1^	6cHS	[31]
*Danio rerio* nitrobindin(II)	5cHS:Soret band at 430 nm; 6cLS:shoulder at 417 nm	5cHs: ν_3_ 1473 cm^−1^; ν_2_ 1561 cm^−1^; ν_10_ 1606 cm^−1^ 6cLS: ν_3_ 1498 cm^−1^; ν_2_ 1585 cm^−1^ ν_10_ overlapped to vinyl stretching modes (ν_C = C_)	Mixture of 5cHS and 6cLS	[50]

## 2. Globins

Globins are a superfamily of globular heme-binding proteins, including several members with a wide range of functions [51,52]. The two most prominent members are tetrameric Hb and monomeric Mb, which are often used as macromolecular models for structural and functional studies. However, globin genes have been discovered in almost every genome, ranging from prokaryotes to higher eukaryote organisms. Of note, novel globin family members were recently found in vertebrates, e.g., neuroglobin (Ngb) in neuronal and glial cells [53,54] and cytoglobin (Cygb) in fibroblasts [55], endowed with peculiar functions.

From a structural point of view, in all 3/3 α-helical globins, the heme group is located between the A-B-E α-helices on one side and the F-G-H α-helices on the other side with the fifth coordination bond of the Fe atom formed by the proximal HisF8 residue [56,57,58]. On the contrary, the 2/2 fold is composed of four α-helices (B, E, G, and H), which create a bundle around the heme group (B/E and G/H antiparallel pairs connected by a loop) [59] (Figure 2). Interestingly, several internal cavities were recently discovered, which play an important role in orchestrating the diffusion of ligands and reshaping the heme pocket [60,61,62].

As for the functional activity of globins, in addition to O_2_ transport and storage, they also display pseudo-enzymatic properties, including peroxidase activity, NO/O_2_ metabolism, fatty acids metabolism, gas sensing, antiapoptotic and antiproliferative effects, cytoprotection in neuronal and cancer cells, and antimicrobial and anti-inflammatory functions [65]. The binding kinetics for the main ligands (O_2_, CO, and NO) to globins have been investigated and fully characterized during the past decades (Table 2).

The functional properties of globins can be modulated both allosterically and covalently [66]. The noncovalent regulation implies the transient variation of ligand-binding properties due to the interaction with a second ligand at the level of a distinct binding site [61,67,68]. Instead, covalent regulation implies the long-lasting presence of the modified species (e.g., phosphorylated) [69]. Basically, the way globins are functionally regulated reflects the specific type of modulating signal: a quick transition between two conformations in response to rapidly changing conditions requires an allosteric modulation, whereas a stable modulation signal requires a covalent modification [70].

**Table 2 biomolecules-13-00683-t002:** Rate constants of ligand binding to globins.

Protein	Ligand	*k*_on_ (μM^−1^s^−1^)	*k*_off_ (s^−1^)	Reference
Human Hb (R-state, pH = 7, 20–25 °C)	O_2_ CO NO	66 6 60	20 0.008 0.00003	[71] [71] [71]
Sperm whale Mb (pH = 7, 20 °C)	O_2_ CO NO	14 0.5 17	12 0.02 0.00012	[72] [72] [73]
Human Ngb (pH = 7, 25 °C)	O_2_ CO NO	140 38 150	0.8 0.007 0.0002	[74] [75] [76]
Human Cygb (pH = 7, 20 °C)	O_2_ CO	30 5.6	0.35 0.003	[75] [75]

### 2.1. Hemoglobin

Hb is broadly recognized as a paradigm of cooperativity and allosteric modulation in proteins whose main function in vertebrates is oxygen binding and transport from the lung to tissues. However, Hb has been extensively studied and characterized during the last decades, unveiling its behavior as a polyfunctional molecule endowed with catalytic activity (e.g., nitrite reductase, NO dioxygenase, monooxygenase, alkylhydroperoxidase, esterase, lipoxygenase) and involved in several physiological processes, such as nitric oxide metabolism, metabolic reprogramming, pH regulation, and redox balance maintenance [70]. Structurally, Hb is a tetramer consisting of two α-subunits (α1, α2) and two β-subunits (β1, β2). The two αβ dimers (α1β1, α2β2) are arranged around a two-fold symmetry axis, resulting in the central water cavity whose shape is influenced by the Hb state. Indeed, O_2_ binding and release from Hb has been historically explained based on the two-state allosteric model of Monod, Wyman, and Changeux (MWC) [67] in terms of equilibrium between two states: the tense (T) state (unliganded), which exhibits low affinity for O_2_, and the relaxed (R) state (liganded), which exhibits high affinity for O_2_. This provided a structural explanation for the cooperative effects that facilitate binding and release of O_2_ in vivo [77]. This model assumes equivalence of all four subunits and relies on an equilibrium between the two quaternary conformations. Based on the presence of both chemically different subunits (α and β) and a single two-fold axis of symmetry, extensions of the MWC model have been proposed as well as alternative models, such as the sequential one [78,79] and the Herzfeld–Stanley (HS) model [80]. Later on, Perutz introduced the stereochemical model, which proposes that the breaking of inter-subunit salt bridges promotes the transition to the R state; indeed, oxygen binding to the T state results in an in-plane movement of the heme–Fe accompanied by the movement of the proximal HisF8 and the associated F-helix, which causes salt bridge breaking [81,82,83,84]. Szabo and Karplus further revised this model by including thermodynamic and energetic observations, proposing that low oxygen affinity in the T state mainly arises from the steric repulsion between HisF8 and the porphyrin [85,86]. The most recently proposed model, which highlights differences in the tertiary structure, is the Tertiary Two State (TTS) model, originally described by Henry and coworkers and based on spectroscopic evidence of tertiary transitions [87,88]. This model states that high and low affinity conformations of individual Hb subunits exist in equilibrium within each quaternary structure, which allows incomplete coupling between tertiary and quaternary transitions.

The balance between the T and R states is affected by both endogenous heterotropic ligands (e.g., 2,3-bisphosphoglycerate, protons, carbon dioxide, chloride) and synthetic allosteric effectors that modulate Hb–O_2_ affinity. In hemoglobin, one of the two axial coordination positions of the heme–Fe is occupied by the proximal His, while the other is available for external ligands. Thus, the five-coordinate heme–Fe is characteristic of unliganded Hb; the sixth coordination position is unoccupied, the heme is domed in the direction of the proximal His, and the iron is out of the heme nitrogen plane by ∼0.2–0.4 Å. Oxygen binding occurs via coordination of O_2_ to the vacant axial position, and the heme–Fe becomes six-coordinate. A difference exists between the alpha and beta subunits; the heme group is nearly planar in the alpha but ruffled in the beta. Indeed, the Fe–O(1)–O(2) angles in the alpha and beta subunits are 153 degrees and 159 degrees, respectively. The oxygen molecule forms a hydrogen bond to the N epsilon of HisE7 in the alpha subunit, but either none or a weak one in the beta subunit [89]. Carbon monoxide is a stronger ligand than oxygen. The Fe–CO complexes of Hb are bent with the Fe–CO axis, which is not orthogonal to the heme plane. Nitric oxide displays a 1500-fold higher affinity for the heme–Fe than CO, and the Fe–NO complex of Hb is bent with an angle similar to that observed in the Fe–O_2_ complex. Other ligands of ferrous Hb are isocyanides and nitroso aromatic compounds. The O_2_ affinity of Hb may vary over a large range due to the effect of heterotropic ligands that bind to a different site. These include hydrogen ions, chloride, phosphate and other inorganic anions, organic phosphates (e.g., 2,3-bisphospho-D-glycerate and inositol hexakisphosphate), and carbon dioxide. All physiologically relevant heterotropic ligands lower the Hb oxygen affinity [90]. In addition to endogenous ligands, Hb is also known to interact with several xenobiotics, which are capable of allosterically modulating the equilibrium between oxygenated and deoxygenated Hb. Compounds, such as 5-hydroxymethylfurfural, pyridyl derivatives of vanillin, voxelotor (GBT440), and triazole sulfide, shift the equilibrium to a high-oxygen-affinity Hb and, therefore, have clinical value in treating sickle cell anemia. By contrast, compounds, such as nitroglycerine, amyl nitrite, isosorbide dinitrate, nicorandil, sinitrodil, and efaproxiral (RSR13), decrease the Hb affinity for oxygen and, therefore, are good candidates for the treatment of hypoxic or ischemic conditions [91]. The effects caused by xenobiotic binding are due to the conformational changes that occur in the Hb tetramer (allosteric modulation), which in turn cause sequential changes in the affinity at other heme sites. Eventually, even though Hb is not an enzyme, it has been found to be endowed with catalytic activity for which it has been elected an “honorary enzyme”. Indeed, Hb displays enzymatic behavior as a nitrite reductase with maximum NO generation rates occurring near the R-to-T allosteric structural transition [92].

### 2.2. Myoglobin

In the context of monomeric globins, sperm whale Mb (swMb) represents a good model to describe quickly adapting allosteric modulation by heterotropic effectors [8,93]. swMb acts as a reserve supply of O_2_ and facilitates the O_2_ flux within myocytes. Lactate, a product of glycolysis under anaerobic conditions (e.g., during prolonged physical effort), behaves as a heterotropic allosteric effector, which leads to decreased O_2_ affinity for swMb. Mechanistically, lactate seems to impair the access of O_2_ to the heme distal pocket of swMb by hydrogen bonding to the Arg(Lys)CD3–HisE7–Thr(Val)E10 triad, thus stabilizing the closed HisE7 conformation [93].

### 2.3. Neuroglobin

An example of covalent modulation of globin activity is represented by the human neuroglobin (Ngb). Ngb is the third discovered member of the human globin family. It is a small monomeric, hexa-coordinated hemoprotein expressed in neuronal cells of both central and peripheral nervous systems, which binds several ligands (e.g., O_2_, CO, and NO) and also displays pseudo-enzymatic properties [94]. Ngb plays a crucial role in neuroprotection from hypoxia in vitro and in vivo. Indeed, it undergoes hypoxia-dependent phosphorylation, which affects the coordination of the heme Fe atom and, in turn, the reactivity of the hemoprotein. The kinetics and thermodynamics of the binding between the distal HisE7 and the heme Fe atom are indeed modulated by two different mechanisms, i.e., (i) phosphorylation and (ii) the redox state of the CysCD7/CysD5 residues pair (Figure 3). Ngb phosphorylation by intracellular kinases was shown to increase the nitrite reductase activity of the hemoprotein by three-fold, altering the coordination of the heme Fe atom [95]. On the other hand, the reduction of the CysCD7–CysD5 disulfide bond was shown to reduce the dissociation rate of HisE7 from the hexa-coordinated Fe atom and increase the value of the equilibrium constant for HisE7 binding to the sixth coordination position of the Fe atom by approximately one order of magnitude. In turn, O_2_ affinity decreased to the same extent [74]. Thus, the neuroprotective role of Ngb is regulated by a ligand-linked, slow phosphorylation/dephosphorylation cycle and by intramolecular disulfide bond formation and dissociation. This facilitates heme Fe–ligand binding and enhances the rate of anaerobic nitrite reduction to NO, eventually contributing to the cellular response to hypoxia.

### 2.4. Cytoglobin

A hybrid regulatory mechanism has been described for human cytoglobin (Cygb), which couples the allosteric modulation by lipid binding with the formation and dissociation of an intramolecular disulfide bond. Cygb function is not yet clearly understood; however, it seems to have a role in protecting cells against oxidative stress. Indeed, it has been proposed that it assists O_2_ transport to the mitochondrial respiratory chain, functions as a NO dioxygenase, and facilitates nitrite reduction and NO generation under anaerobic conditions [96,97,98,99,100]. Similarly to Ngb, Cygb displays a bis-histidyl, hexa-coordinated heme Fe atom, so that the cleavage of the distal HisE7–Fe bond is a prerequisite for the binding of exogenous ligands [101,102]. The heme reactivity depends on the lipid (e.g., oleate) binding to the long N- and C-terminal disordered protein extensions, which promotes the transition from a hexa- to penta-coordinated heme Fe atom, thus stabilizing the penta-coordination geometry [103,104]. Moreover, the redox state of the CysB2/CysE9 residue pair in ferrous Cygb directly influences the heme Fe reactivity (Figure 3). The CysE9 and HisE7 residues are located on opposite sides of the E-helix; therefore, the formation of the CysE9–CysB2 bond seems to exert some strain on HisE7 [105]. This conformational change may perturb the local molecular dynamics, thus influencing ligand binding.

**Figure 3 biomolecules-13-00683-f003:**
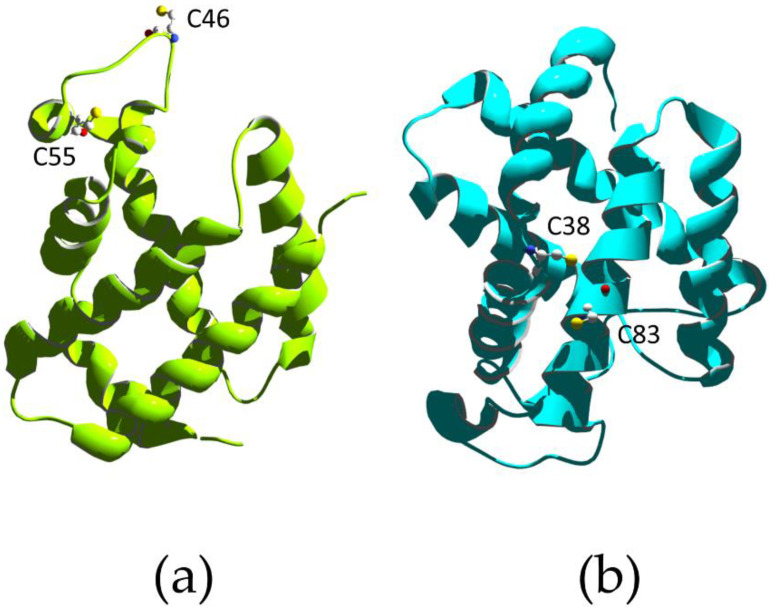
X-ray structure of human neuroglobin (**a**) (PDB ID 1OJ6 [106]) and human cytoglobin (**b**) (PDB ID 2DC3 [103]). The cysteine residues that regulate the heme Fe reactivity are labeled.

## 3. Cytochromes

Cytochromes are ubiquitous, evolutionary-conserved, electron-carrying proteins involved in central redox catalysis. They show characteristic strong absorption of visible light due to the presence of the iron-containing heme prosthetic group. Thus, they are grouped in different classes based on their longest wavelength absorption band in their reduced (Fe^2+^) state: cytochrome a (605 nm), cytochrome b (≈565 nm), and cytochrome c (550 nm). All classes are involved in the mitochondrial respiratory chain. Cytochrome c (cytc) is the most abundant protein in mitochondrial cristae, where it transports electrons from cytochrome c reductase (complex III) to cytochrome c oxidase (complex IV). Its six-coordinated heme iron is, therefore, able to cycle from the reduced form to the oxidized form. This redox property is mainly due to the axial coordination of the M80 sulfur atom (Fe–S bond). Under these conditions, the heme-related reactivity of cytc is negligible in both the reduced and oxidized forms. However, the redox potential of cytc is not only dependent on M80 axial coordination. In this context, Bren and coworkers demonstrated that deformations of metal porphyrins (i.e., ruffling, saddling, doming) affect the redox potentials, electron transfer kinetics, and ligand-binding properties of heme-binding proteins. Ruffling is the main deformation in cytc. Density Functional Theory (DFT) calculations of His/Met-ligated heme demonstrated that, in addition to decreasing the electronic coupling of the cofactor with an external reductant, the ruffling also modified the reduction potential of the heme [107,108].

Upon binding to anionic lipids, cytc acquires peroxidase activity. This implies a conformational change of the protein, which also involves partial unfolding of its tertiary structure [109,110,111]. The protein has to change the ligation state of the heme group and also the spin state of the heme–Fe, which is a pH-dependent process, in order to acquire peroxidase activity [112,113]. Indeed, the native protein with its heme–Fe in the six-coordinate low-spin state with His and Met as axial ligands displays low accessibility of the active site to the solvent and the inability of M80 to accept a proton. Cardiolipin (CL), a glycerophospholipid enriched in the mitochondrial inner membrane, plays a major role in regulating both the binding and function of cytc on the mitochondrial inner membrane, owing to its two phosphate head groups that can strongly interact electrostatically with cytc. Upon binding to CL, cytc acquires heme-related, ligand-binding properties and reactivity similar to that of globins [114,115,116,117,118,119,120]. In particular, CL–cytc acquires the peroxidase activity that is critical for CL redistribution across the intermembrane space [114]. This is crucial for mitochondrial biology since CL in the inner membrane maintains the protein complexes’ structure and function, while its oxidation and subsequent redistribution on the outer membrane allows cytc to be released in the cytosol, thus triggering the apoptotic process [121]. As reported for CO and O_2_, nitric oxide (NO) also binds to the heme, causing conformational changes and allosteric regulation of the activity [56,122,123,124]. Indeed, NO binds to the distal side of the ferrous heme giving rise to hexa-coordinated adducts with His as the axial proximal ligand. Alternatively, NO binding to the heme distal side induces the dissociation of the proximal His residue, leading to penta-coordinated complexes. In its ferric form, CL–cytc can reversibly bind NO to form the CL–cytc(III)–NO complex, which undergoes reductive nitrosylation to form the CL–cytc(II)–NO complex [116]. It has been demonstrated that CL binding induces a change in the cytc tertiary structure that leads to a rearrangement of K72, K73, and K79 with a consequent cleavage of the M80 Fe–S bond, thus leading to a five-coordinated species with globin-like reactivity [15,125].

Recent advances in bacterial genome sequencing revealed many genes encoding novel c-type cytochromes, which contain multiple heme cofactors [126]. One example is the *Geobacter sulfurreducens*, whose genome contains 111 *cytc* genes, and 73 of them display two or more heme groups (one containing 27 heme groups). This huge number of multi-heme cytochromes helps these bacteria survive in energy-limited environments by utilizing a diverse range of respiratory pathways and terminal electron acceptors, including metal ions. In particular, extracellular electron transport allows some bacterial species to use solid minerals as electron sources. The electron uptake process involves integral outer-membrane, multi-heme cytochromes that contain at least four heme groups, which are essential to form a direct electron-transport pathway [127]. The activity of multi-heme cytochromes is not limited to electron transfer with several proteins able to bind substrates at one of the heme groups and display enzymatic activity, such as the pentaheme cytochrome c nitrite reductase. The latter is a homodimer; the protein folds into one domain composed by three long alpha-helical segments, two of which are key elements for dimerization. Within each monomer, the five heme groups form both near-parallel and near-perpendicular arrangements. The active site is located at heme 1, which displays an unusual lysine coordination arising from the presence of a CXXCK motif in the protein sequence, which results in an amine nitrogen as the proximal ligand. The distal ligand varies across species but has been observed as water/hydroxide, sulfate, and azide; the latter two being inhibitors of nitrite reductase activity [128,129,130] (Figure 4).

Multi-heme, cytochrome-mediated extracellular electron transfer is being pursued for wiring bacteria to electrodes in bio-electrochemical renewable energy technologies, especially in the context of fuel-to-electricity and electricity-to-bioproduct conversions. Emerging topics include the role of multi-heme cytochromes in interspecies electron transfer and in supporting the design of novel protein-based bioelectronic components [132].

## 4. Hemopexin

The glycoprotein hemopexin is the plasma protein that displays the highest binding affinity for heme. It is mainly expressed in the liver as an acute-phase protein upon inflammation, and it is the major carrier of heme in the plasma, thus preventing heme-mediated oxidative stress and heme-bound iron loss [133]. After heme binding, hemopexin undergoes a conformational change allowing for interaction with the CD91 receptor, mainly expressed on hepatocytes, followed by internalization. Inside the cell, heme is catabolized while the hemopexin-receptor complex is recycled [134]. Thus, the most important physiological role of hemopexin is to act as an antioxidant after blood heme overload, which is critical to cope with severe pathological situations such as sepsis [135].

Human hemopexin consists of a single 439-aminoacid polypeptide chain with six intrachain disulfide bridges [136]. The protein comprises two homologous domains of approximately 200 residues each joined by a 20-aa linker. These domains have a unique four-bladed, β-propeller fold that is a variant of the larger β-propeller domains found in heterotrimeric G proteins, clathrin, and integrins, which play a crucial role in protein–protein interactions. The N-terminal threonine residue is blocked by an O-linked galactosamine oligosaccharide, and the protein has five glucosamine oligosaccharides N-linked to the acceptor sequence Asn–X–Ser/Thr. A unique feature of hemopexin is the high content of tryptophan residues (i.e., eighteen residues arranged in four clusters). The heme binds between the two propeller domains in a pocket bounded by the interdomain linker. Two histidine residues coordinate the heme iron, His213 from the linker and His266 from the C-terminal domain, giving a stable bis-histidyl Fe(III) complex (Figure 5).

A body of evidence suggests that heme-bound plasma proteins may display ligand binding and (pseudo-) enzymatic properties [19]. In particular, ferrous hexa-coordinated heme–hemopexin (HPX–heme(II)) binds CO, NO, and cyanide by detaching the H213 nitrogen–iron coordination bond [19]. O_2_ binds transiently and is followed by fast HPX–heme(II) oxidation [45,138].

CO reversibly binds to penta-coordinated rabbit HPX–heme(II) (at pH < 7.0) with a second-order rate constant *k*_on_ = 1.9 × 10^3^ M^−1^s^−1^ and a first-order rate constant *k*_off_ = 5.0 × 10^−4^ s^−1^. These values reflect both slower carbonylation and decarbonylation kinetics compared to other penta-coordinated hemoproteins [19]. On the other hand, at pH > 7.0, CO binds to hexa-coordinated rabbit HPX–heme(II) with a second-order rate constant *k*_on_ = 2.1 × 10^2^ M^−1^s^−1^ and a first-order rate constant *k*_off_ = 5.0 × 10^−4^ s^−1^, the latter not being affected by the protonation state of H213.

NO binds reversibly to rabbit HPX–heme(II) with a second-order rate constant *k*_on_ = 6.3 × 10^3^ M^−1^s^−1^ and a first-order rate constant *k*_off_ = 9.1 × 10^−4^ s^−1^ [138]. NO also binds to ferric HPX–heme(III) with a second-order rate constant *k*_on_ = 1.3 × 10^1^ M^−1^s^−1^ and a first-order rate constant *k*_off_ < 1.0 × 10^−4^ s^−1^. The binding is followed by reductive nitrosylation [139].

Eventually, O_2_ and peroxynitrite are reported to react with HPX–heme(II)–NO yielding HPX–heme(III) and NO_3_^−^_,_ thus conferring pseudo-enzymatic properties to HPX–heme(II)–NO [139,140].

## 5. Human Serum Heme–Albumin

HSA displays an extraordinary ligand binding capacity, providing a depot and carrier for many endogenous and exogenous compounds [21]. HSA influences the pharmacokinetics and pharmacodynamics of several drugs (e.g., antibiotics, anticoagulants, antineoplastics, antivirals, anesthetics, anxiolytics, and nonsteroidal anti-inflammatory drugs) [141]. Heme–HSA has been characterized by several spectroscopic techniques, such as optical [142], NMR [23], Raman and resonance Raman [48], EPR [45], and X-ray absorption spectroscopy (XAS) [143] (Figure 6). As reported in Table 1, the spectroscopic characterization of Fe(III)heme–HSA highlights a 5cHS heme and a weak Fe(III)–O_Tyr_ coordination in agreement with the remarkable Fe(III)–O distance (2.73 Å) determined from the crystal structure.

Although HSA is a globular, monomeric protein, its multidomain structural organization resembles that of multimeric proteins [144]. Therefore, the heme-binding thermodynamics and reactivity are regulated by heterotropic interactions as well as by ligand competition [145]. Specific binding of drugs to Sudlow’s site I can allosterically decrease heme-binding affinity to its site and vice-versa [146]. The allosteric mechanisms have been observed for several drugs: warfarin [147]; imatinib [148]; the anti-HIV drugs abacavir, atazanavir, didanosine, efavirenz, emtricitabine, lamivudine, nelfinavir, nevirapine, ritonavir, saquinavir, stavudine, and zidovudine [149]; isoniazid and rifampicin [150]; flavonoids [151]; Δ9-tetrahydrocannabinol [152]; apomorphine [153]; and bezafibrate and clofibrate [49]. Some drugs display an allosteric effect by binding to secondary sites as observed for ibuprofen [154] and benserazide [153]. Eventually, some drugs can competitively displace heme from its HSA binding site as observed for cantharidin, retinoic acid, and retinol [155,156].

Ferrous heme–HSA (Fe(II)heme–HSA) displays globin-like reactivity [157,158,159]. Fe(II)heme–HSA has been demonstrated to be a mixture of several species, mainly a four-coordinated intermediate-spin species, a five-coordinated high-spin species, and a six-coordinated low-spin species, with different abundances at different pH values [160]. Association and dissociation constants for Fe(II)–NO binding in Fe(II)heme–HSA were reported by Coletta and coworkers [157]. The second-order rate constant for Fe(II)heme–HSA nitrosylation was observed to decrease by one order of magnitude in the presence of a saturating concentration of ibuprofen and warfarin [157]. In addition, the denitrosylation kinetics of Fe(II)heme–HSA were affected by ibuprofen [158] and warfarin [159]. Table 3 reports NO association and dissociation constants to Fe(II)heme–HSA in the absence of drugs and in the presence of warfarin and ibuprofen. Both drugs decrease the second-order nitrosylation reaction rate by one order of magnitude with a concomitant effect on the first-order denitrosylation rate. Warfarin binds primarily to Sudlow’s site I that is functionally linked to the modulatory site FA2, which in turn regulates HSA conformation [142,145,146]. However, ibuprofen binds primarily to Sudlow’s site II, which is not functionally linked to the heme site, indicating that ibuprofen may bind at three different sites with thermodynamic dissociation constant values of 3.1 × 10^−7^ M, 1.7 × 10^−4^ M, and 2.2 × 10^−3^ M [158]. NO also binds reversibly to Fe(III)heme–HSA to form a transient Fe(III)–NO complex followed by dismutation to Fe(II)–NO^+^. This species undergoes nucleophilic attack by OH^−^ (at pH > 6.5) with the formation of Fe(II)–NO [161].

Fe(III)heme–HSA also catalyzes peroxynitrite isomerization to nitrate (*k*_on_ = 4.1 × 10^5^ M^−1^s^−1^), thus preventing nitration of free added tyrosine [162]. In the presence of ibuprofen, isoniazid, rifampicin, chlorpropamide, digitoxin, furosemide, indomethacin, phenylbutazone, sulfisoxazole, tolbutamide, and warfarin, the reaction is allosterically impaired with a reaction kinetics superimposable to that observed in the absence of Fe(III)heme–HSA [150,162,163].

## 6. Nitrobindin

In 2010, structural studies on the 166-residue protein coded by *Arabidopsis thaliana* (*At*) gene locus At1g79260.1 revealed specific and high affinity binding to heme with the ferric derivative of this heme protein binding NO. The protein was named nitrobindin (Nb) due to its structural and functional similarity to nitrophorins (NPs), a group of lipocalins involved in NO transport [27]. The compact barrels of both proteins have a large hydrophobic cavity with the heme iron coordinated by a proximal histidine and the end opposite of the heme-binding site enclosed by a 3_10_ helix. However, Nb forms a compact ten-stranded, β-barrel structure (whereas NPs contain eight β-strands) and a more open entry to the heme pocket with the heme positioned at the rim of the cavity and well exposed to the solvent. Structural crystallographic analysis of *At*–Nb shows that the metal center of the heme is coordinated by His158 and, contrarily to globins, no distal histidine was identified as the sixth ligand [27]. However, at the opposite side of the heme plane, an additional histidine (His76) is found to be parallel to it, although located too far away from the heme–Fe atom to coordinate it (with a distance of 6.9 Å calculated on PDB ID 3EMM using CCDC Mercury software). Multiple van der Waals interactions contribute to the stabilization of the binding with the heme. The cavity where the heme is located is delimited by highly hydrophobic residues (Phe44, Met75, Thr98, Leu100, Val128, Ile131, Met148, and Leu159) placed within 4 Å from it. A heme propionate forms hydrogen bond interactions with Thr40, while the other propionate is exposed to the solvent, and no close contacts with any aminoacidic side chains are found [29]. Furthermore, while NPs feature an Asp residue in the distal pocket to stabilize the protonated primary amine of the bound histamine, Glu78 is present in the distal pocket of *At*–Nb but does not seem to stabilize bound imidazole (Figure 7).

Homologs of *At*–Nb were identified and structurally characterized later on, such as a human homolog containing the thanatos-associated protein 4 (THAP4) domain (*Hs*–THAP4) and two structural homologs in *Mycobacterium tuberculosis* (*Mt*–Nb), suggesting that nitrobindins could represent a novel family of all β-barrel, heme-binding proteins present in prokaryotes and eukaryotes. A bioinformatics investigation based on the amino acid sequences and 3D structures of *At*–Nb and *Hs*–THAP4 showed that all Nb-like proteins from *bacteria* to *H. sapiens* show conservation of the His158 residue coordinating the heme–Fe atom and the His76 side chain in contact with the heme moiety on the distal side [28].

Kinetic studies of NO binding to *Mt*–Nb(III), *Hs*–Nb(III), and *At*–Nb(III) revealed similar values of ^NO^*k*_on_ for NO binding in agreement with those reported for nitrosylation of *Rhodnius prolixus* nitrophorin *Rp*–NP1(III) and *Rp*–NP4(III) but lower than *Rp*–NP2(III) and *Rp*–NP3(III). A much lower value was determined for *Physeter catodon* myoglobin *Pc*–Mb(III) due to a remarkable solvent inaccessibility to the heme in comparison with the corresponding ferric Nbs. In accordance, denitrosylation kinetic studies assessed that the ^NO^*k*_off_ values of *At*–Nb(III), *Mt*–Nb(III)–NO, *Hs*–Nb(III)–NO, and *Rp*–NP2(III-NO match well with those of *Rp*–NP3(III)–NO but are higher than *Rp*–NP1(III)–NO, *Rp*–NP4(III)–NO, and *Pc*–Mb(III) [27,30,164,165]. The variability in NO dissociation rates could be related to differences in the stabilization of the NO–Fe(III) bond controlled by different hydrogen bond interactions.

Despite the structural similarities between *Rp*–NPs and Nbs, the affinity of histamine for *Mt*–Nb(III), *At*–Nb(III), and *Hs*–Nb(III) was found to be seven to eight orders of magnitude lower than *Rp*–NPs. A docking simulation showed that the side chains of His85 and Ser87 in *Mt*–Nb(III), Ser97, and Thr98 in *At*–Nb(III), and Thr91 and Asn90 in *Hs*–Nb(III) sterically encumber the binding of the heme groups’ Fe (III) to histamine [30,166,167]. More recently, the kinetic and spectroscopic features of nitrosylation of ferric and ferrous nitrobindin from *Danio rerio* zebrafish were found to match those of the other Nbs previously investigated. Such nitrosylation kinetics are associated with a high free-energy barrier for NO association to the heme (~38.3 ± 1.0 kJ/mol), which may be due to the crowded network of water molecules that shields the heme distal pocket in Nbs. A different trend is observed in the NO dissociation process with dissipation of the distal water molecules’ wall and the more open heme pocket of Nbs leading to a lower energy barrier [50,168]. This is reflected in a much faster NO exchange rate than other hemoproteins (such as myoglobins), electing nitrobindins as regulators of NO levels in the blood.

With nitric oxide playing a key role in disease resistance in plants, such as *Arabidopsis thaliana*, ferric *At*–Nb was hypothesized to be involved in wounding and pathogenic infections by transporting and rapidly releasing NO at the infection site, therefore, contributing to the generation of reactive oxygen species (ROS) as NO may reduce O_2_ to superoxide radicals that can dismute to H_2_O_2_ [27,120,121]. The inability of *At*–Nb to establish stable complexes with O_2_, CO, or NO in the presence of air excludes possible roles in O_2_ storage or transport.

Over the past years, De Simone et al. [30] characterized the structure and potential physiological function of nitrobindins through multiple in vitro assays and cell survival analyses and highlighted their evolutionary conserved function role in NO sensing and scavenging of reactive nitrogen species (RNS) and ROS as components of pools of antioxidant protein systems.

An investigation of the effect of *Mt*–Nb(III) on the peroxynitrite-induced nitration of L-tyrosine demonstrated its protective role over L-tyrosine nitration, while the process is not prevented when it comes to *Mt*–apo–Nb. The ability of ferric *Mt*–Nb to detoxify peroxynitrite was confirmed in vivo upon transformation in the BL21 (DE3) *Escherichia coli* strain, subsequently treated for 24 h with increasing concentrations of the peroxynitrite generator 3-morpholinosydnonimine (SIN-1), demonstrating how it ensures bacteria growth also in the presence of RNS [30]. The survival of *M. tuberculosis* in the host suggests ferric Nb as part of a detoxifcation system scavenging RNS and ROS produced by the immunity response, as reported for mammalian globins [169,170] and mycobacterial truncated hemoglobins [171]. If confirmed with future additional in vitro and in vivo experiments, Nb may emerge as a therapeutic target for the treatment of not only pulmonary tuberculosis but also extra-pulmonary manifestations, such as ocular infections involving conjunctiva, cornea, and sclera [172].

*Hs*–Nb could act as a sensor of peroxynitrite levels in which peroxynitrite binding to the C-terminal *Hs*–Nb domain of THAP4 may regulate the transcriptional activity residing at the N-terminal domain. In vitro tests on HEK293 cells ectopically overexpressing *Hs*–Nb suggested that *Hs*–Nb, which localizes mainly in the cytoplasm and partially in the nucleus, may play a role in the detoxification process of human cells from peroxynitrite in a similar way to *Mt*–Nb as protein nitration upon treatment with SIN-1 was detected to be reduced by 40% and 60% after 2 h and 24 h, respectively. In addition, significantly increased cell viability was observed for HEK293 cells overexpressing *Hs*–Nb after a 2 h treatment with SIN-1 with respect to untransfected HEK293 cells and HEK293 transfected with an empty vector [30].

Finally, the recently observed nitrosylation dynamics of *Dr*–Nb confirm the importance of a potential role of Nbs as a regulator of NO levels in poorly oxygenated tissues, such as the fishes’ eye retina, which is subjected to high hydrostatic pressure under deep diving conditions [50,168].

## Figures and Tables

**Figure 1 biomolecules-13-00683-f001:**
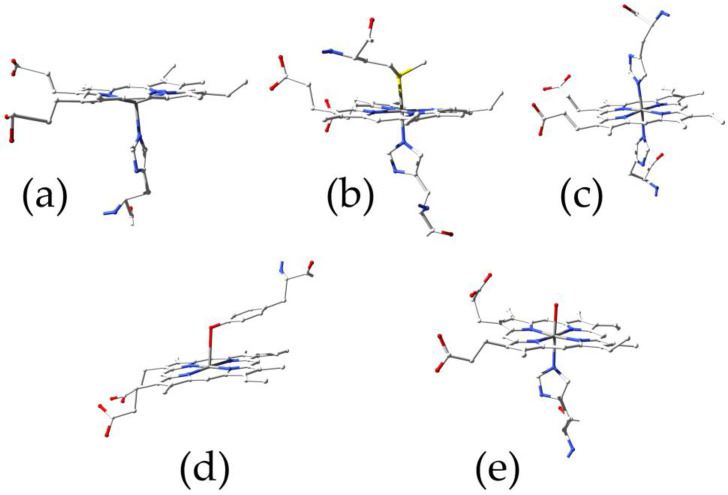
Principal heme coordination modes: (**a**) globins (PDB ID 4HHB); (**b**) cytochromes (PDB ID 3ZCF); (**c**) hemopexin (PDB ID 1QHU); (**d**) serum albumin (PDB ID 1O9X); (**e**) nitrobindin (PDB ID 6R3W).

**Figure 2 biomolecules-13-00683-f002:**
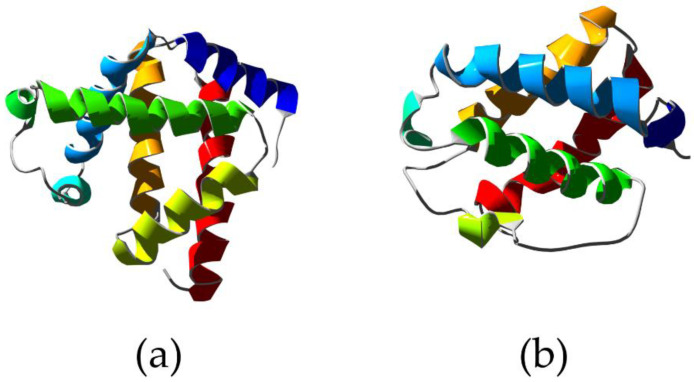
X-ray structure of *Physeter catodon* myoglobin showing the typical 3/3 alpha helical sandwich (**a**) (PDB ID 1A6K [63]) compared to the structure of truncated hemoglobin from *Chlamydomonas eugametos* showing the 2/2 alpha helical sandwich (**b**) (PDB ID 1DLY [64]). Ribbons are colored following secondary structure succession.

**Figure 4 biomolecules-13-00683-f004:**
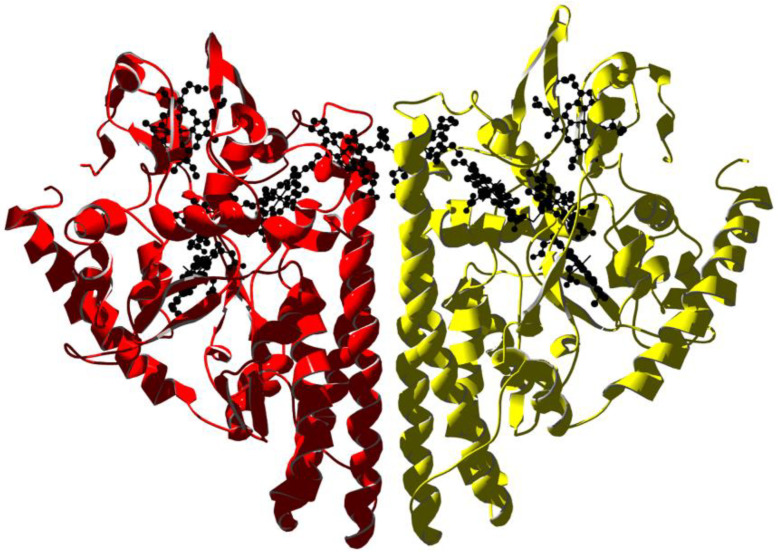
X-ray structure of the periplasmic pentaheme cytochrome c nitrite reductase from *E. coli* (PDB ID 1GU6 [131]). The homodimer is rendered with subunits in yellow and red. Heme molecules are in black.

**Figure 5 biomolecules-13-00683-f005:**
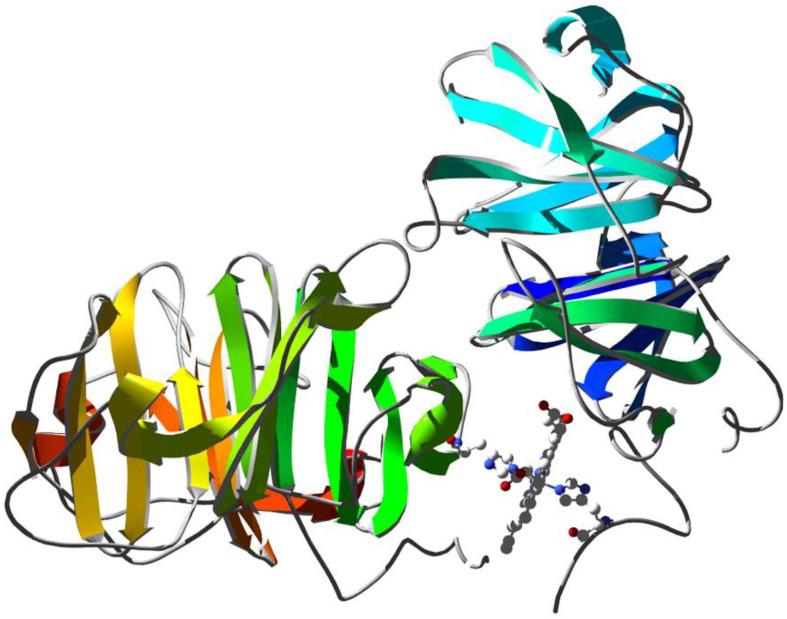
X-ray structure of rabbit serum hemopexin deglycosylated and complexed with heme (PDB ID 1QHU [137]). Ribbons are colored following secondary structure succession. Heme is shown together with the two histidine residues that coordinate the Fe ion.

**Figure 6 biomolecules-13-00683-f006:**
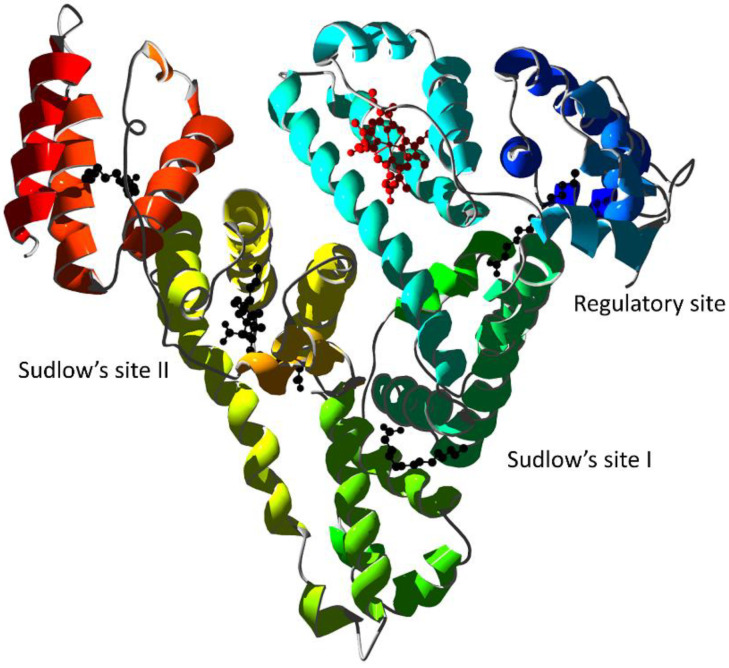
X-ray structure of human serum albumin complexed with ferric heme and myristic acid (PDB ID 1N5U [24]). Ribbons are colored following secondary structure succession. Heme is rendered in red and myristate ions as black ball-and-sticks. The regulatory site and the Sudlow’s drug binding sites are indicated.

**Figure 7 biomolecules-13-00683-f007:**
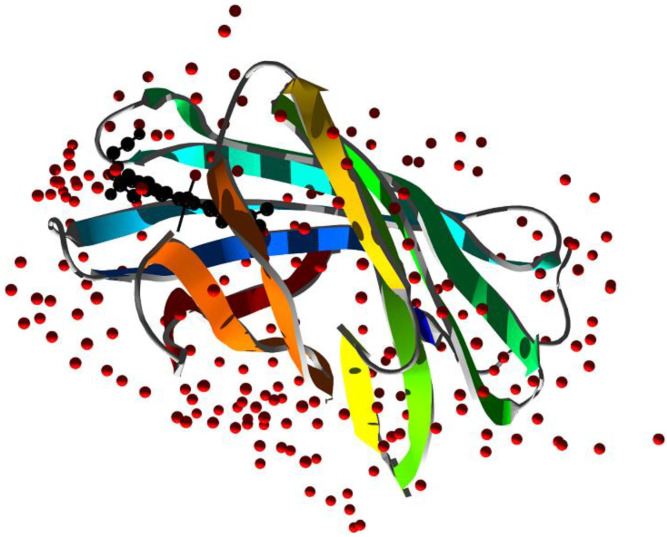
X-ray structure of *M. tuberculosis* nitrobindin (PDB ID 6R3W [30]). Ribbons are colored following secondary structure succession. Heme is rendered in black and water molecules as red spheres.

**Table 3 biomolecules-13-00683-t003:** Rate constants for nitrosylation (*k*_on_) and denitrosylation (*k*_off_) of ferrous heme–HSA.

Drug	*k*_on_ (M^−1^s^−1^)	*k*_off_ (s^−1^)
No drug	6.3 × 10^6^ [157]	1.4 × 10^−4^ [158]
Ibuprofen	4.1 × 10^5^ [157]	9.5 × 10^−3^ [158]
Warfarin	4.8 × 10^5^ [157]	8.6 × 10^−4^ [159]

## Data Availability

Not applicable.

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
