# Peer review of "Ligand-Based Regulation of Dynamics and Reactivity of Hemoproteins"

_biomolecules, 2023, doi:10.3390/biom13040683_

Round 1

Reviewer 1 Report

This review article covers a range of heme binding proteins: globins, cytochromes, hemopexin, human serum albumin and nitrobindin. The focus is both on ligands that bind to the heme or to other allosteric sites. It provides an interesting and useful overview of the nature of allostery. The review primarily highlights work from the Fasano and Ascenzi labs, which given their productivity in this area is probably fine. It will provide a useful primer for those looking for an introduction to the diversity of heme-based proteins in biology. The paper has only one figure, which given the diversity heme proteins discussed and the focus on allostery is a bit of a weakness.

-In the globin section, a figure comparing the 3/3 and 2/2 globin folds would be useful.

-In the globin section, a figure of neuroglobin and/or cytoglobin showing the disulfide bonds that may regulate ligand binding affinity would be useful.

-In the cytochrome section, perhaps a figure of a selection of multiheme cytochromes. The pentaheme cytochrome c nitrite reductase with the unusual CXXCK motif would be useful perhaps with inhibitor bound.

- In the hemopexin section, a figure that highlights some of the structural features described in the text would be useful.

-Similarly, for Human Serum Albumin, a figure that shows heme bound and highlights the binding sites mentioned in the text (Sudlow’s site I, Sudlow’s site II, modulatory site FA2) would be useful.

-In the nitrobindin section, a figure that perhaps highlights the water structure in the heme binding pocket could be useful, given its potential effect on kinetics of ligand binding.

Minor points:

-line 233, p. 5 “alfa-helix” should be “alpha-helix”

-p. 7 line 312, “Coletta and coworkers [ref].” The numerical citation needs to be provided.

-p. 7 line 315, Table X should be Table 1.

-p. 9 line 391, I don’t think “eliging” is a word in English.

-p. 9 line 399, the numerical reference needs to be provided for De Simone et al. [ref].

-p. 13, lines 649-650, for ref 99, there is no year provided.

Author Response

We are thankful to the Reviewer for these suggestions. With the present review article, we would like to contribute to the Special Issue that celebrates Massimo Coletta’s pivotal work in this field. For this reason, we mainly focused on (and cited) his work and that of his collaborators.

To meet reviewer’s requests, we added at least one figure per section, namely Figure 2 (X-ray structure of Physeter catodon myoglobin showing the typical 3/3 alpha helical sandwich (a; PDB ID: 1A6K [63]), compared to the structure of truncated hemoglobin from Chlamydomonas eugametos showing the 2/2 alpha helical sandwich (b; PDB ID 1DLY [64]). Ribbons are colored following secondary structure succession), Figure 3 (X-ray structure of human neuroglobin (a; PDB ID: 1OJ6 [105]) and human cytoglobin (b; PDB ID 2DC3 [102]). The cysteine residues that regulate the heme Fe reactivity are labeled), Figure 4 (X-ray structure of the periplasmic pentaheme cytochrome c nitrite reductase from E. coli (PDB ID: 1GU6 [130]). The homodimer is rendered with subunits in yellow and red. Heme molecules are in black), Figure 5 (X-ray structure of rabbit serum hemopexin deglycosylated and complexed with heme (PDB ID: 1QHU [136]). Ribbons are colored following secondary structure succession. Heme is shown together with the two histidine residues that coordinate the Fe ion), Figure 6 (X-ray structure of human serum albumin complexed with ferric heme and myristic acid (PDB ID: 1N5U [24]). Ribbons are colored following secondary structure succession. Heme is rendered in red and myristate ions as black ball-and-sticks. The regulatory site and the Sudlow’s drug binding sites are indicated), and Figure 7 (X-ray structure of M. tuberculosis nitrobindin (PDB ID: 6R3W [30]). Ribbons are colored following secondary structure succession. Heme is rendered in black and water molecules as red spheres).

We also addressed all minor issues/typos in the revised manuscript.

All changes are tracked in the revised manuscript.

Reviewer 2 Report

The authors summarize the structures and functions of several heme-binding proteins, including globins, cytochromes, hemopexin, and nitrobindins. However, there are many other types of globins, such as gas sensor proteins, and cytochromes also have many other different proteins, such as cytochrome P450 and NOS. It would be helpful if the authors could expand their review article to make it more comprehensive.

However, the review article only scratches the surface of the topic, and I do not see a well-organized summary of spectroscopic studies or how ligands modulate enzyme reactivity. The authors should make more efforts to provide a clear picture. It would be beneficial to create a table summarizing the UV-vis spectra, EPR, and Raman spectra of each protein described in the article. Additionally, including a clear crystal structure showing key residues that play a crucial role in regulating enzyme activity would be useful. Finally, providing a table showing all kinetic data when binding O2, NO, or CO would be helpful.

Author Response

We are thankful to the Reviewer for these suggestions. We are aware that the present review article does not represent a comprehensive overview of heme-binding proteins dynamics and reactivity. However, our aim was to specifically highlight Massimo Coletta’s work in this field, since this was the topic of the Special Issue. For this reason, we mainly discussed his work and that of his collaborators, selecting those hemoproteins that have been thoroughly investigated by them.

To meet reviewer’s suggestions and concerns, and to improve readability, we added in the revised version of the manuscript six figures (Figures 2 – 7) and two tables (Table 1 “Main spectroscopic properties of representatives of the hemoprotein classes described in this review” and Table 2 “Rate constants of ligand binding to globins”), in order to give the reader more insights into binding and structural dynamics of the selected hemoproteins. Please notice that Table 1 does not include EPR parameters as unligated Fe(II) hemoproteins have a very short electron relaxation time and spectra are difficult to observe. Most papers refer to NO complexes which are very informative due to the quadrupolar coupling. However, our choice was not to include the EPR characterization of these species as it would require an extensive discussion, which is out of the scope of this review.

All changes are tracked in the revised manuscript.

Round 2

Reviewer 2 Report

The authors have addressed my major concern, it could be published at present form. 

Author Response

We are grateful to Reviewer 1 for helpful suggestions.